# Controlled Synthesis of Au Nanocrystals-Metal Selenide Hybrid Nanostructures toward Plasmon-Enhanced Photoelectrochemical Energy Conversion

**DOI:** 10.3390/nano10030564

**Published:** 2020-03-20

**Authors:** Ling Tang, Shan Liang, Jian-Bo Li, Dou Zhang, Wen-Bo Chen, Zhong-Jian Yang, Si Xiao, Qu-Quan Wang

**Affiliations:** 1Department of Physics, Hunan Normal University, Changsha 410081, China; tl1924411950@163.com (L.T.); cwbdeemail@126.com (W.-B.C.); 2Key Laboratory for Matter Microstructure and Function of Hunan Province, Hunan Normal University, Changsha 410081, China; 3Institute of Mathematics and Physics, Central South University of Forestry and Technology, Changsha 410004, China; 4School of Materials Science and Energy Engineering, Foshan University, Foshan 528000, Guangdong Province, China; 5Hunan Key Laboratory of Super Microstructure and Ultrafast Process, School of Physics and Electronics, Central South University, Changsha 410083, China; Douzhang@csu.edu.cn (D.Z.); zjyang@csu.edu.cn (Z.-J.Y.); sixiao@csu.edu.cn (S.X.); 6Department of Physics, Key Laboratory of Artificial Micro-and Nano-Structures of the Ministry of Education, Wuhan University, Wuhan 430072, China; qqwang@whu.edu.cn

**Keywords:** gold-metal selenide, hollow hybrid nanostructure, surface plasmon resonance, morphology manipulation, photoelectrochemical response

## Abstract

A simple method for the controllable synthesis of Au nanocrystals–metal selenide hybrid nanostructures via amino acid guiding strategy is proposed. The results show that the symmetric overgrowth mode of PbSe shells on Au nanorods can be precisely manipulated by only adjusting the initial concentration of Pb^2+^. The shape of Au–PbSe hybrids can evolve from dumbbell-like to yolk-shell. Interestingly, the plasmonic absorption enhancement could be tuned by the symmetry of these hybrid nanostructures. This provides an effective pathway for maneuvering plasmon-induced energy transfer in metal–semiconductor hybrids. In addition, the photoactivities of Au–PbSe nanorods sensitized TiO_2_ electrodes have been further evaluated. Owing to the synergism between effective plasmonic enhancement effect and efficient interfacial charge transfer in these hybrid nanostructures, the Au–PbSe yolk-shell nanorods exhibit an outstanding photocurrent activity. Their photocurrent density is 4.38 times larger than that of Au–PbSe dumbbell-like nanorods under light irradiation at *λ* > 600 nm. As a versatile method, the proposed strategy can also be employed to synthesize other metal–selenide hybrid nanostructures (such as Au–CdSe, Au–Bi_2_Se_3_ and Au–CuSe).

## 1. Introduction

Nowadays, nanocrystals (NCs) with multiple compositions and heterointerfaces have received great attention due to promising multifunctional and synergistic physicochemical properties [1,2,3,4,5,6,7]. As a promising example, metal–semiconductor hybrid NCs, which integrate plasmon metals and semiconductors into one nanosystem, exhibit intriguing properties and functionalities far beyond those of the corresponding individual counterparts [8,9,10,11,12,13,14,15,16]. For instance, local surface plasmon resonance (SPR) from metal NCs can be tailored by changing their effective dielectric environment [10,17]. Electromagnetic field amplification or resonant energy transfer can be employed to improve the light absorption [18,19], photoluminescence [20,21], and optical nonlinear effects of hybrid nanosystems via plasmon–exciton interaction [22,23]. Additionally, a Schottky barrier will be formed when a semiconductor is brought into contact with a metal. At here, the metal can also act as an electron trap for photoelectrons from the semiconductor after excitations, improving the charge carrier separation and reducing the recombination [24,25,26]. Recently, the plasmon-induced hot-electron transfer from metal nanostructures into strong coupled semiconductors has been reported, providing a new paradigm for solar energy conversion [16,27,28,29].

By virtue of narrow bandgap and favorable band position, (Au nanorods (NRs))-based metal-chalcogenide plasmonic hybrids (such as Au–CdS, Au–Cu_2-*x*_Se, Au–AgCdSe hybrid NCs, etc.) have increasingly became a research hotspot in the fields of photoelectric and photocatalysis [30,31,32,33,34]. Many efforts have also been made to investigate the physical and chemical properties of the hybrid NCs [2,35,36,37,38], and the results indicated that their properties are highly dependent on the size, shape, composition, and spatial distribution of each component [24,30,39,40,41]. Therefore, a large number of schemes keep emerging towards synthesis of these hybrid nanostructures under the joint endeavours of the majority of scientific researchers [42,43,44,45,46]. Among them, seed-mediated selective growth of semiconductor shells based on metal nanocrystals has unique advantages, owing to abundant and controllable plasmon properties from the designed metal nanostructures [33,44,47,48]. As a typical structure, Au NRs possess transverse SPR (T-SPR) and longitudinal SPR (L-SPR) absorption bands, which can be tuned from visible (Vis) to near-infrared (NIR) regions depending on the dimensions of the NRs [49,50,51]. Especially, based on the plasmon–exciton interactions, (Au NRs)-selenide semiconductor hybrid NCs and has been synthesized successfully, which provides many feasible schemes for manipulating their optical, photocatalytic and photoelectric properties [31,32,33,48,52]. For instance, Au–Cu_2−*x*_Se NCs have been synthesized by Zou’s group via a selenium-mediated, in which an interesting dual-plasmon phenomenon has been observed experimentally [44]. Zhang and co-workers proposed a cation-exchange scheme to obtain the dumbbell-like Au–CdSe NRs, which possess highly efficient photoelectrochemical hydrogen generation beyond visible region in these NRs [33]. In our previous works, we synthesized AgCdSe NCs by selectively growing at the surface of Au NRs, and obtained symmetric and asymmetric Au–AgCdSe hybrid nanorods, offering an effective pathway for tuning their optical properties [48]. However, the controlling of hetero-nanostructures is still a challenging job, especially for manipulation the symmetry of hybrid nanocrystals using the same materials.

In this paper, Au nanocrystals–metal selenide hybrid nanostructures with controllable shells were firstly prepared by the amino acid guidance strategy. Taking Au–PbSe hybrid NRs as an example, the growth kinetics of hybrid nanostructures have further been investigated. Meanwhile, we also explored the influences of symmetry modes of the nanoshells on optical and photoelectrochemical properties of the hybrid structures. As a general method, the strategy proposed could be successfully used to synthesize other metal–semiconductor hybrid nanostructures, such as Au–PbSe, Au–CdSe, Au–Bi_2_Se_3_ and Au–CuSe hybrid nanostructures.

## 2. Experimental

### 2.1. Chemicals and Materials

Chloroauric acid (HAuCl_4_·H_2_O, 99.99%), silver nitrate (AgNO_3_, 99.8%), glycine acid (Gly, 99.5%), cetyltrimethylammonium-bromide (CTAB, 99.0%), sodium borohydride (NaBH_4_, 96%), L-ascorbic acid (AA, 99.7%), sodium hydrate (NaOH, 96.0%), hydrochloric acid (HCl, 36~38%), tetrabutyl titanate (C_16_H_36_O_4_Ti, 98%), sodium sulfite anhydrous (Na_2_SO_3_, 97%), sodium sulfide nonahydrate (Na_2_S·9H_2_O, 98%), selenium dioxide (SeO_2_, 99.0%), lead acetate (Pb(C_2_H_3_O_2_)_2_·3H_2_O, 99.0%), copper chloride (CuCl_2_, 99.0%), bismuth nitrate (Bi(NO_3_)_3_·5H_2_O, 99%) and cadmium acetate (Cd(C_2_H_3_O_2_)_2_·3H_2_O, 99%) were purchased from Sinopharm Chemical Reagent Co. Ltd. (Shanghai, China). All chemicals were used as received and without further purification. All aqueous solutions were freshly prepared in the water obtained by ultra-pure water system from ZOOMWO Co. Ltd. (Hunan, China).

### 2.2. Growth of Au NRs

Au NRs were prepared by using a seed-mediated growth method [51]. Au seed solution was obtained by adding 0.2 mL of ice-cooled NaBH_4_ solution (10 mM) into a 5 mL aqueous solution containing HAuCl_4_ (5 mM) and CTAB (200 mM). The Au seed solution was kept at room temperature with evenly stirring for 1 h before we use it. To grow Au NRs, 1 mL of 5 mM HAuCl_4_ and 0.01 mL of 0.1 M AgNO_3_ were first mixed with 6 mL of 0.2 M CTAB. Then, 0.015 mL of 0.1 M HCl was added, followed by the addition of 0.7 mL of 0.01 M AA. After the solution was mixed by inversion, followed by the addition of 0.1 mL of the CTAB-stabilized Au seed solution. The resultant solution was gently mixed for 10 s and was left undisturbed overnight, and then the obtained Au NRs were collected by centrifugation and then redispersed in 0.2 M CTAB aqueous solution for further use.

### 2.3. Growth of Au@Se NRs

Au@Se nanorods with opened ends were prepared by a one-pot method. In brief, 1 mL of the as-prepared Au nanorods and 0.05 mL of 0.1 M AA were firstly mixed in a 10 mL round-bottom test tube. After stirring for 3 min, 0.02 mL of 0.1 M SeO_2_ was added. The mixed solution was then incubated at 38 °C with evenly stirring for 5 h. Finally, Au@Se nanorod seeds were obtained. At this stage, the color of solution changed from dark red to celadon. The obtained samples were then centrifuged at 9500 rpm for 9 min, and then redispersed in 1 mL of 0.2 M CTAB solution for further use.

### 2.4. Synthesis of Au–PbSe Hybrid NRs

In a typical procedure, Gly[Pb^2+^] that acted as the Pb^2+^ precursor was prepared by 0.02 mL of 0.1 M Pb(C_2_H_3_O_2_)_2_·3H_2_O and 1.0 mL of 0.2 M Gly in a 10 mL round-bottom test tube, followed by the addition of 1 mL of the as-prepared Au@Se NRs seed. The pH of the reaction solution was regulated to 9.8 by injecting 2 M NaOH solution. Subsequently, 0.02 mL of 0.1 M AA was added and kept at 80 °C under vigorously stirring for 90 min. The obtained products were centrifuged at 8000 rpm for 9 min, and then redispersed in water solution. The obtained sample was Au–PbSe yolk-shell hybrid NRs.

### 2.5. Preparation of (PbSe NCs)@TiO_2_, (Au–PbSe Dumbbell NRs)@TiO_2_ and (Au–PbSe Yolk-Shell NRs)@TiO_2_ NWs Film Electrodes

Firstly, TiO_2_ nanowires (NWs) film was grown on fluorine-doped tin oxide (FTO) glass substrate by the hydrothermal method. 7.5 mL of HCl was diluted with 7.5 mL ultrapure water and mixed with 0.25 mL C_16_H_36_O_4_Ti in a 50 mL beaker. After stirring for 30 min, this white suspension mixture and a clean FTO glass substrate were transferred to a 25 mL Teflon-lined stainless-steel autoclave, where the FTO substrate was tilted 45 degrees in the solution. The sealed autoclave was heated in an electric oven at 150 °C for 5 h, and then let it cool down to room temperature slowly. A white TiO_2_ NWs film was uniformly deposited on the FTO glass substrate, which was thoroughly washed with ultrapure water and then dried at 60 °C in a vacuum. The sample was annealed in air at 550 °C for 3 h to improve the crystallinity of TiO_2_ NWs. Subsequently, to obtain TiO_2_ NWs electrodes sensitized with Au–PbSe yolk-shell hybrid NRs, the adsorbent was prepared by adding 10 mg of Au–PbSe NRs into 1 mL of 0.17 M alcohol solution. The obtained TiO_2_ NWs film was immersed into it for 3 min, then was thoroughly washed with ethanol and dried at 50 °C in a vacuum. After repeated the adsorption process three times, the expected (Au–PbSe yolk-shell NRs)@TiO_2_ NWs film electrode was obtained. (PbSe NCs)@TiO_2_ and (Au–PbSe dumbbell NRs)@TiO_2_ NWs electrodes were obtained by the same way, excepting using the equal amounts of counterparts instead of Au–PbSe yolk-shell NRs in the adsorption process.

### 2.6. Characterizations

The transmission electron microscopy (TEM), high-angle annular dark-field scanning TEM (HAADF-STEM) and high-resolution TEM (HRTEM) observations were performed with a FEI Tecnai F20 transmission electron microscope operated at 200 kV. Energy dispersive X-ray spectroscopy (EDS) analyses were performed on an EDS incorporated in the HRTEM (FEI Tecnai F20, Hillsboro, OR, USA). The scanning electron microscopy (SEM) images were collected with a field-emission SEM (FEI NovasSEM-450, Hillsboro, OR, USA). The X-ray diffraction (XRD) analyses were performed on a Bruker D8-advance X-ray diffractometer (Bruker AXS, Billerica, Germany) with Cu kα irradiation (*λ* = 1.5406 Å). Absorption spectra of the samples were measured using a TU-1810 UV-Vis spectrophotometer (Purkinje General Instrument Co. Ltd., Beijing, China).

### 2.7. Photocurrent Response and Incident Photon to Current Conversion Efficiency (IPCE) Measurements

TiO_2_ NWs film was fabricated into photoanodes by soldering a copper wire onto a bare part of the FTO substrate. The working electrode area is 1 cm^2^. Na_2_SO_3_ (0.25 M) and Na_2_S (0.35 M) aqueous solution (pH ≈ 12), Ag/AgCl, and Pt were employed as electrolyte, reference electrode, and counter electrode, respectively. Photocurrent experiments were carried out at a scan rate of 10 mV/s under the solar simulator (Beijing SOFN photoelectric instruments Co. Ltd., 7IS0503A, Beijing, China) using a 350 W xenon lamp equipped with an air mass 1.5 global (AM 1.5G) filter as illumination source (100 mW/cm^2^). After adding a 600 nm long-pass filter to AM 1.5G simulated sunlight, the light irradiation power change to 3.8 mW/cm^2^. The IPCE spectra were measured by using a 150 W Xenon lamp equipped with a monochromator.

## 3. Results and Discussion

### 3.1. Controllable Synthesis of Au–PbSe Hybrid NRs

In this work, Au NRs are chosen as our initial materials because of their fascinating optical properties. A schematic outline of the amino acid guidance strategy is shown in Figure 1. The preparation progress can be divided into two steps. Firstly, an amorphous Se shell is deposited on the Au NRs by reducing SeO_2_ with AA. When CTAB is used as the stabilizer and morphology inducer, the end-opened Au@Se core@shell NRs can be obtained owing to the anisotropy of NR and the strong affinity between Au and Se. In order to further obtain metal selenide nanoshells, Au@Se NRs are used as seeds and Se sources in the next step. Previous literatures have revealed that amino acids can interact with metal clusters through the amine and carboxylic groups [53], Gly molecules can selectively adsorb on two ends of Au@Se NRs by the anchoring bonds. In addition, Gly can be also combined with metal ions to form a complex. Therefore, Gly can act as a transporter in the reaction, carrying metal ions from the solution to the end regions of Au NRs. Subsequently, Se atoms in the shells turned into Se^2−^ and Se^4+^ ions by disproportionation reaction. The solution is adjusted to an appropriate pH value with NaOH aqueous solution, when Se^2−^ and metal ions encounter in the solution, metal selenide nanocrystals are expected to form and further facilitate the disproportionation process of Se. In addition, Se^4+^ ions will be reduced back to Se^0^ again through AA in the solution and continue to participate in the above reaction. As such, the Au–selenide hybrid NRs with hollow shells are obtained. Furthermore, the symmetry manipulating of semiconductor hollow shells can be realized by changing the initial concentration of the added cation precursors. The major reactions involved in this process can be summarized as the following [54,55]:(1)$$3{\mathrm{Se}}_{\left(s\right)}+3H_{2}O\to 2{\mathrm{Se}}_{\left(\mathrm{aq}\right)}^{2-}+{\mathrm{SeO}}_{3\left(\mathrm{aq}\right)}^{2-}+6H_{\left(\mathrm{aq}\right)}^{+}$$
(2)$${\mathrm{nSe}}_{\left(\mathrm{aq}\right)}^{2-}+2M_{\left(\mathrm{aq}\right)}^{n+}\to M_{2}{\mathrm{Se}}_{n\left(s\right)}$$
(3)$$C_{6}H_{8}O_{6\left(\mathrm{aq}\right)}+{\mathrm{SeO}}_{3\left(\mathrm{aq}\right)}^{2-}+2H_{\left(\mathrm{aq}\right)}^{+}\to C_{6}H_{4}O_{6\left(\mathrm{aq}\right)}+{\mathrm{Se}}_{\left(s\right)}+2H_{2}O$$

Taking Au–PbSe hybrid NRs as an example, the morphology and structure characterizations at each growth stage are characterized in detail by TEM images in Figure 2a–c. The initial Au NRs synthesized by a seed-mediated growth method exhibit regular rod shape, the corresponding average length and width are about 56 and 16 nm, respectively. After the addition of SeO_2_ and AA, incubating at 38 °C for 5 h, the designed Au@Se core@shell NRs seeds are shown in Figure 2b. As we can see, when the added amount of Se source is 20 μL of 0.1 M, the average thickness of Se shell is about 22 nm. Subsequently, as 20 μL of 0.1 M Pb^2+^ precursor is added, the closed Au–PbSe hybrid NRs with transversely symmetrical hollow shells are achieved. The corresponding TEM images shown in Figure 2c reveal that Au NRs are embedded on the longitudinal axis of hollow ellipsoidal shells. There is a 12 nm gap between the Au core and the external PbSe and the semiconductor nanoshells of about 13.8 nm shells are wrapped around the NRs, which is a typical yolk-shell nanostructure. Interestingly, the Au NRs yolk keeps close contacts with the semiconductor shells at both ends, which provides an easy access for the interfacial electron interactions. In order to obtain more crystallographic details, XRD analyses were performed for the samples at different reaction stages. As shown in Figure 2d, the diffraction peaks from Au can be well identified according to joint committee on powder diffraction standards (JCPDS) cards no. 04-0784. The result indicates that the crystalline phase of the initial Au NRs is cubic. After coating Se shells on the sides of Au cores, no new diffraction peaks are found from the samples, indicating that the Se shells are amorphous. As for Au–PbSe hybrid NRs, a new XRD pattern emerges, which can be indexed as the cubic PbSe phase (JCPDS card no. 06-0354). Figure 2e is HRTEM image of the individual hybrid NR at the interface of Au NR yolk and PbSe shell. As expected, a lattice plane spacing of 0.306 nm in the shell region is observed, which agrees well with the (200) lattice planes of the cubic PbSe crystal. Also, the lattice plane distance of 0.235 nm in the yolk region corresponds to the (111) planes of face-centered cubic (fcc) Au. To further confirm the composition of the samples, the corresponding EDS is exhibited in Figure 2f, the spectrum shows strong Au, Pb, and Se peaks in addition to the C, O, and Cu peaks generated by the copper grid. This result demonstrates the existences of Au, Pb, and Se elements in the yolk-shell nanostructures. Additionally, the HAADF-STEM imaging and EDS elemental mapping of single nanoparticle are displayed in Figure 2g–j. As can be seen, Pb (red) and Se (cyan) are uniformly distributed in the shell region, and the yolk is composed of Au (blue), demonstrating that the well-defined Au–PbSe yolk-shell hybrid nanostructures are obtained in our experiments.

### 3.2. Growth Kinetics and Symmetry Controlling of Au–PbSe Hybrid NRs

We monitored the evolution of shape for the products obtained at different reaction stages with corresponding TEM images in Figure 3. Figure 3a exhibits the initial Au@Se NRs with a uniform nanoshell. It is notable that Se atoms mainly distribute in the shell region, while Pb^2+^ ions disperse in the solution. The diffusion flux of Se^2−^ ion is obviously higher than that of Pb^2+^ ion at the reaction interface. After reaction for 5 min, some small voids appear in the inner of shells owing to the Kirkendall effect [56,57]. As displayed in Figure 3b. Meanwhile, a set of distinct lattice fringes emerge on the outer layer of the shell, which indicates that the PbSe NCs with cubic phase are beginning to form. Continue to reaction for 25 min, Figure 3c shows the Kirkendall voids coalesce into a bigger one via the fast surface diffusion of Se atoms. With the reaction proceeding, the expected Au–PbSe yolk-shell hybrid nanostructure is obtained at 60 min, as shown in Figure 3d. What’s more, two ends of the NRs are wrapped by PbSe NCs at the initial stage, it can be attributed to the selectively adsorption of Gly[Pb^2+^] complexes on the ends of Au NRs. Their corresponding low magnification TEM images are displayed in Appendix A. As such, it is reasonable to assume that the structures of hybrid NRs are determined by the combined actions of Kirkendall effect and Gly guiding mechanism in our experiments.

To further test this hypothesis, the diffusion flux of precursors at the reaction interface can be tuned by their initial concentration in this experiment. Here, there are four parallel samples with the same seed of Au@Se NRs, and the related initial molar concentrations Pb^2+^ (*C*(Pb^2+^)) are set to 0.25 mM, 0.5 mM, 0.75 mM and 1.0 mM, respectively. Corresponding TEM images of the Au–PbSe hybrid nanostructures are exhibited in Figure 4a–d. At a low Pb^2+^ ion concentration, most of Pb^2+^ ions in the solution are transported to both ends of the Au@Se NRs in the forms of Gly[Pb^2+^] complexes. Meanwhile, Se atoms in the shells are reduced to Se^2−^ ions by AA and diffused into the solution under the action of concentration gradient force. Once Se^2−^ ions encounter Pb^2+^ ions at the end regions of Au NRs, PbSe NCs will be formed. When *C*(Pb^2+^) = 0.25 mM, owing to the binding of Gly to metal ions, the diffusion rate of Se^2−^ ion is higher than that of Pb^2+^ ion in the end regions of NRs, which inhibit the formation of Kirkendall voids. Hence, the Gly-guiding mechanism plays a dominant role in the growth of PbSe at this moment. As shown in Figure 4a, Au–PbSe NRs with two porous PbSe NCs on the ends of NRs are obtained, in which the semiconductor shell has a longitudinally symmetrical distribution. Here, we call them as Au–PbSe porous-dumbbell NRs. With increasing the concentration of Pb^2+^ in the reaction solution, the diffusion rate of Pb^2+^ ions increases. As *C*(Pb^2+^) increases to 0.5 mM, the diffusion flux of Pb^2+^ ions is larger than that of Se^2-^ ions at the reaction interface. As shown in Figure 4b, the Au–PbSe nanodumbbells with longitudinally symmetrical hollow shells are observed and called as hollow-dumbbell NRs, which implies that two mechanisms of Kirkendall effect and Gly-guiding metal cation transport exist simultaneously in the growth process. In addition, the increase of Pb^2+^ diffusion rate also causes the reaction interface to move towards Se shell area. As expected, when *C*(Pb^2+^) increases to 0.75 mM, umbrella-shape PbSe NC gradually forms and then curls toward the sides of NR, the corresponding TEM image is displayed in Figure 4c. Finally, under the joint action of these two mechanisms, the Au–PbSe yolk-shell nanostructure with transversely symmetric hollow shell can be obtained at *C*(Pb^2+^) = 1 mM. The results are illustrated in Figure 4d. The HAADF-STEM imaging in conjunction with EDS elemental mapping in Figure 4e further reveal the coexistence of Au NRs and PbSe cap regions for hollow nanodumbbells. As thus, the symmetry of the hollow PbSe shells on Au NRs can be manipulated from longitudinally to transversely symmetric distributions by simply adjusting the initial Pb^2+^ concentration, which is ascribed to the combined actions of Kirkendall effect and the glycine-guiding cation transport. In addition, as exhibited in Appendix A, Au–PbSe yolk-shell hybrid NRs with different shell thicknesses and nanovoid sizes are also achieved by adjusting Se shell of Au@Se nanoseeds.

As a general method, our strategy can be extended to the synthesis of other metal-selenide hybrid nanostructures. By simply replacing the Gly[Pb^2+^] complexes with corresponding cation precursors and regulating the pH value of the system, the Au–CdSe, Au–Bi_2_Se_3_ and Au–CuSe hybrid nanostructures can be successfully obtained. The detailed operation processes are described in the Appendix A. The morphology and detailed crystallographic structures of the corresponding hybrid NRs are shown in Appendix A. As we can see, all the prepared hybrid nanostructures have uniform rod-like Au cores, and Au NRs are wrapped with semiconductor nanoshells. More interestingly, two ends of Au NRs are clearly enclosed with semiconductor counterparts in three samples. The results further confirm the guiding roles of Gly in the growth of hybrid nanostructures. Furthermore, Au–PbSe yolk-shell nanocubes and triangular nanoprisms can also be achieved by the similar method, as shown in the Appendix A. Compared with the other strategies (i.e., cation exchange, overgrowth assisted with Ag wetting layer), the above growth scheme of metal–selenide hybrid nanostructures based on Se nanoshells and amino acid guiding strategy obviously possesses some advantages such as good crystallinity, controllable shell morphology and no introduction of other impurities (such as Cu^2+^ or Ag^+^).

### 3.3. Optical Properties of Au–PbSe NRs with Different Shells Distributions

It is known that the optical properties of metal–semiconductor plasmonic nanostructures are closely related to the distribution of semiconductor component. The controllable morphology of Au–PbSe hybrid NRs can be used to regulate their SPR properties. As see in Figure 5, the SPR peaks of original Au NRs are observed at 518 nm and 752 nm, respectively. After 13 nm Se shell is coated, the corresponding peaks of Au@Se NRs, respectively, shift to 578 nm and 1018 nm, accompanying with the emergence of a new transverse resonance mode at 680 nm. Those can be attributed to the high refractive index of the Se layer.

Subsequently, for porous (*C*(Pb^2+^) = 0.25 mM) and hollow (*C*(Pb^2+^) = 0.5 mM) Au–PbSe nanodumbbells, their T-SPR peaks keep almost unchanged in comparison to the original Au NRs, because semiconductor component is mainly distributed at both ends of the NRs. As *C*(Pb^2+^) is adjusted from 0.5 to 0.75 mM, the hollow semiconductor components change into umbrella-shaped caps at both ends of the NRs. Their corresponding L-SPR peak exhibits a slightly blue shift and the T-SPR peak shows a red shift, owning to the formation of a new surrounding environment. Meanwhile, the hollow nanodumbbells shows a significantly enhanced absorption peak in the NIR region, which might be due to the strong longitudinal coupling between plasmons and excitons in the hybrids. When the closed hollow shells with transverse symmetry are formed (*C*(Pb^2+^) = 1 mM), the L-SPR peak of Au–PbSe yolk-shell NRs redshifts to 1450 nm. Most interestingly, a significant enhancement of T-SPR band at visible region is displayed in the yolk-shell structures, it might be ascribed to the strong transverse coupling between plasmons and excitons in the hybrids. The results provide an effective pathway for maneuvering optical properties of metal–semiconductor hybrids by the distribution of hollow nanoshells. The local field confinements and enhancements of the metal–semiconductor hybrids with different symmetric hollow shells are further revealed by using FDTD simulations in Appendix A. The L-SPR and T-SPR wavelengths of the starting Au NRs are located at 780 nm and 509 nm, respectively. After the PbSe hollow shells were grown on the Au NRs, the dumbbell nanostructure exhibits corresponding T-SPR enhancement, and the core-shell structures exhibit L-SPR enhancement. Their variation tendency is in accord with the above experimental results. Three kinds of plasmonic nanostructures exhibit different field enhancement distributions, indicating that the symmetry of shells affects greatly the optical properties of the hybrid NRs. Here, the Au–PbSe yolk-shell hybrid NRs with transverse shells have a strong T-SPR band in the visible region, besides considerable longitudinal resonance absorption. As thus, the broad SPR absorption bands from Au–PbSe NRs, ranged from Vis to NIR region, match well with the peak wavelength of solar spectrum, allowing for the potential solar applications.

### 3.4. Photoelectrochemical Performances Tests

The photoelectrochemical tests of TiO_2_ NWs photoanodes sensitized by Au–PbSe hybrid NRs with different symmetry were further performed in a three-electrode cell, with Ag/AgCl as the reference electrode and a Pt wire as the counter electrode. In comparison, TiO_2_ NWs electrodes are sensitized with hollow PbSe NCs, Au–PbSe dumbbell-like and yolk-shell NRs, which are labeled as (PbSe NCs)@TiO_2_, (Au–PbSe dumbbell NRs)@TiO_2_ and (Au–PbSe yolk-shell NRs)@TiO_2_, respectively. All sensitizers had the same mass of corresponding components as described in the Appendix A. TEM images of the sensitizers and corresponding extinction spectra of the sensitized TiO_2_ electrodes are shown in Appendix A respectively. The typical SEM images of (Au–PbSe NRs)@TiO_2_ electrode are shown in Appendix A. As shown in Figure 6a, it displays the I-t curves for four parallel electrodes under white-light (AM 1.5G, 100 mW/cm^2^). A clear photo-switching behavior is observed, and the photocurrent of all the sensitized electrodes are enhanced compared with that of TiO_2_ NWs substrate, which implies that there exists an effective photoelectron transfer channel from sensitizer to TiO_2_. The stable photocurrents of 0.49, 0.46, and 0.63 mA/cm^2^ were correspondingly obtained for PbSe@TiO_2_, (Au–PbSe dumbbell NRs)@TiO_2_ and (Au–PbSe yolk-shell NRs)@TiO_2_ NWs electrode. The results suggest that the photocurrent response is obviously dependent on the shell symmetry for metal–semiconductor hybrid nanostructures sensitized electrode. The corresponding photocurrent density-applied potential characteristics are presented in Appendix A.

To further explore the effect of Au–PbSe NRs with different symmetrical shells on the photoactivities of their sensitized TiO_2_ electrodes, Figure 6b shows the I–t curves under light irradiation at *λ* > 600 nm. As seen, no obvious photocurrent response is detected in the pure TiO_2_ NWs substrate under the same testing conditions. After sensitized with PbSe NCs, a stable photocurrent with a density of 12.25 μA/cm^2^ can be observed, which indicates that there is an effective photoelectron transfer process from the PbSe NCs to the TiO_2_ NWs. More interestingly, the intensity of the photoelectric signal is decreased to 10.62 μA/cm^2^ for (Au–PbSe dumbbell NRs)@TiO_2_ electrode. By contrast, the photocurrent density of (Au–PbSe yolk-shell NRs)@TiO_2_ is 4.38 times that.

To understand the charge-transfer process, electrochemical impedance spectroscopy (EIS) experiments were performed at open circuit potential under light illumination (*λ* > 600 nm). Here, *R*_s_ denotes the contact resistances of the electrochemical device, CPE denotes the capacitance phase element, and *R*_ct_ denotes the interfacial charge transfer resistance. As shown in Figure 6c, the values of *R*_ct_ are calculated to be 13.35, 6.74, 2.88 and 1.81 KΩ for TiO_2_, (PbSe NCs)@TiO_2_, (Au–PbSe dumbbell NRs)@TiO_2_ and (Au–PbSe dumbbell NRs)@TiO_2_ electrodes, respectively. *R*_ct_ of (PbSe NCs)@TiO_2_ electrode is significantly smaller than that of pure TiO_2_ electrode. This indicates that there exists a favorable energy level alignment between TiO_2_ NWs and hollow PbSe NCs. Among them, (Au–PbSe yolk-shell NRs)@TiO_2_ has the smallest radius in comparison to the other three electrodes. Therefore, it manifests the fastest charge carriers transfer rate. The physics behind this result can be explained by the highly efficient photoexcitation and electron-hole separation efficiency resulting from the unique spatial architecture of Au–PbSe yolk-shell nanostructures. As for dumbbell-like Au–PbSe NRs, 27.5 nm PbSe NCs are symmetrically distributed on both ends of Au NRs (see in Appendix A). Because of the quantum size effect, it will induce the changes of the energy band structures of PbSe NCs and the formation of unfavorable energy level alignment in the PbSe-TiO_2_ heterojunction. As a consequence, the transfer of electrons from PbSe to TiO_2_ electrode was greatly inhibited. (Au–PbSe dumbbell NRs)@TiO_2_ electrode exhibits the higher interfacial charge transfer resistance than the other sensitized electrodes. These results further confirm the importance of structure and interface optimization for the photoelectric conversion of metal–semiconductor hybrid nanomaterials, which are highly consistent with the above photocurrent tests. To verify whether the observed photoactivity enhancement of (Au–PbSe yolk-shell NRs)@TiO_2_ electrode comes from the plasmon excitation of the hybrid NRs, IPCE for (PbSe NCs)@TiO_2_ and (Au–PbSe yolk-shell NRs)@TiO_2_ electrodes were also performed. Two electrodes are subjected to an incident light with wavelength 600–1000 nm. IPCE values at specific wavelengths are calculated are shown in the Appendix A. As presented in Figure 6d, the measured IPCE action spectrum of the (Au–PbSe yolk-shell NRs)@TiO_2_ electrode matches well with its corresponding SPR absorption spectrum. Additionally, different from (PbSe NCs)@TiO_2_, (Au–PbSe yolk-shell NRs)@TiO_2_ exhibits two evident IPCE enhancement bands located at 670 and 980 nm, which suggests that Au NRs and PbSe nanoshells in the yolk-shell NRs can synergistically promote the photoelectron excitation due to the effective plasmon–exciton interaction under Vis–NIR light (*λ* > 600 nm) irradiation.

To understand the enhanced mechanism of photocurrent response in the sensitized TiO_2_ NWs photoanodes, the processes of inner charge transferring are further discussed in Figure 7. As we know, apart from the irradiance on the electrode, the photocurrent response depends on the light-harvesting efficiency, the yield for electron injection, and the charge collection efficiency. In our research, since the electron scavenger is absent in the electrolyte and sufficiently positive potential, the photocurrent is mainly determined by the light-harvesting efficiency and the yield of electron injection. Firstly, the Au–PbSe yolk-shell hybrid NRs with transverse hollow shells possess a stronger T-SPR band in the visible region than the other plasmonic nanostructures, co-operated with the large optical coupling interface, which are more benefit for the light harvesting and utilization in the visible bands. Secondly, a proper energy level alignment at heterojunction between TiO_2_ and PbSe can also be achieved by adjusting the proper PbSe shell thickness due to the strong quantum confinement effects, which facilitates the electron transfer from PbSe to TiO_2_ [58]. Meanwhile, considering the Fermi level of Au and the Schottky barrier at the Au–PbSe interface, the hot electrons (1.26 eV) derived from the excitation of SPR of Au NRs in Au–PbSe NRs have sufficient energy to surmount the Schottky barrier (≈0.37 eV), so they are injected into the conduction band of PbSe shell [59]. Furthermore, after the hot electrons pass through the PbSe nanobridge at the end of hybrid NRs, they will be drained into the TiO_2_ electrode, and transmit to counter electrode through the extra circuit to drive water-splitting reactions. In addition, the hollow and porous nanoshells not only maintain the independence of the plasmon nanostructures, but also provide a channel for the connection between the Au core and the solution. It facilitates hot holes rapidly eliminating by the hole scavengers and recycle the photoreduction/oxidation reactions. PbSe as a fascinating semiconductor material with a large exciton Bohr radius, high index of refraction and multiple-exciton generation [60,61], more interesting studies are under way on the synergistic properties of hollow Au–PbSe hybrid nanostructures.

## 4. Conclusions

In summary, we have proposed a facile and common route to controlled synthesize Au–PbSe hybrid nanostructures. Based on the as-prepared Au–PbSe NRs, we have investigated the growth kinetics of hybrid nanostructures. Under the combined action of Kirkendall effect and the Gly-guiding cation transport, morphology modulation from longitudinally (dumbbell-like) to transversely (yolk-shell) symmetric overgrowth of shells on Au NRs are achieved, by simply adjusting the initial Pb^2+^ concentration. Interestingly, the plasmonic absorption enhancement with different resonance modes could be tuned by the symmetric distribution of shell on Au NR in our experiments. That provides an effective pathway for maneuvering plasmon–exciton in metal–semiconductor hybrids. In addition, shell-symmetry-dependent photoelectrochemical properties were also observed in Au–PbSe hybrid NRs sensitized TiO_2_ electrodes. The TiO_2_ NWs electrode sensitized with transversely symmetric Au–PbSe yolk-shell NRs exhibited superior photoactivity over the others, which is attributed to the synergism between effective plasmonic enhanced sunlight harvesting and highly efficient interfacial charge transfer in the hybrid nanostructure. As a versatile method, employed Au@Se nanostructures and Gly[M*^n^*^+^] (M*^n^*^+^ = Cd^2+^, Bi^3+^ and Cu^2+^) as reaction precursors, the well-defined Au–CdSe, Au–Bi_2_Se_3_ and Au–CuSe hybrid nanostructures were further obtained with our amino acid guiding strategy. We believe that this study could be extended to the fabrication and functionalization of multicomponent nanostructures with controllable morphology and component distribution. Our findings would boost the development of materials for solar energy conversion.

## Figures and Tables

**Figure 1 nanomaterials-10-00564-f001:**
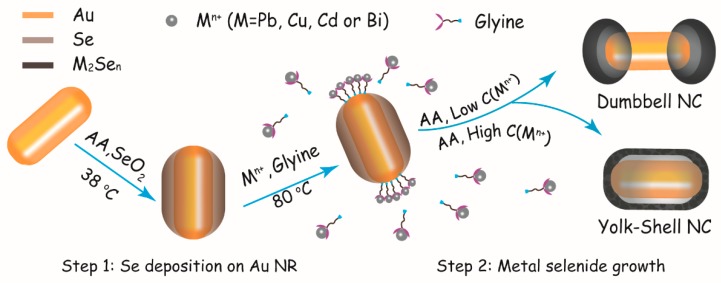
Schematic illustration of controlled growth of Au–selenide hybrid nanocrystals (NCs) with nanovoids. The symmetric distribution of nanoshells on Au nanorods (NRs) can be tuned from longitudinal (dumbbell-like) to transverse (yolk-shell) modes by simply adjusting the initial concentration of metal ion precursors.

**Figure 2 nanomaterials-10-00564-f002:**
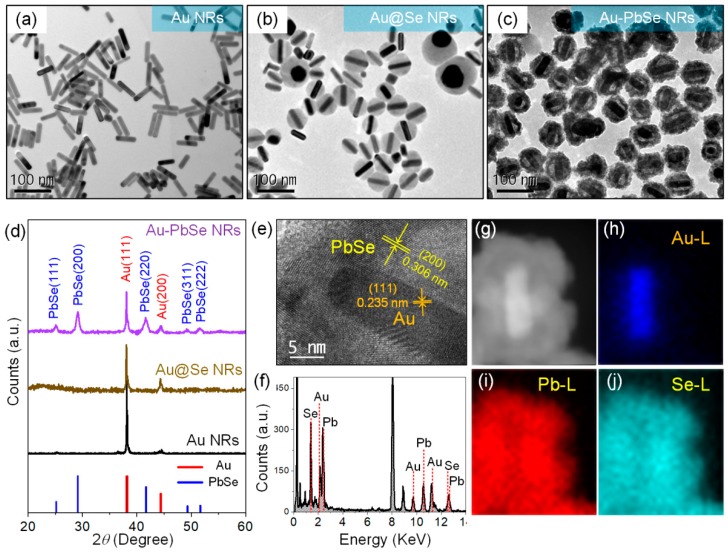
Transmission electron microscopy (TEM) images of (**a**) initial Au NRs; (**b**) Au@Se NRs; (**c**) Au–PbSe yolk-shell NRs; (**d**) X-ray diffraction (XRD) spectra of the samples obtained at different reaction stages; (**e**) high-resolution TEM (HRTEM) image; (**f**) the corresponding energy dispersive X-ray spectroscopy (EDS) analysis; (**g**) high-angle annular dark-field scanning STEM (HAADF-STEM) image of the individual Au–PbSe nanoparticle and (**h**–**j**) EDS elemental maps of Au, Pb and Se, respectively.

**Figure 3 nanomaterials-10-00564-f003:**
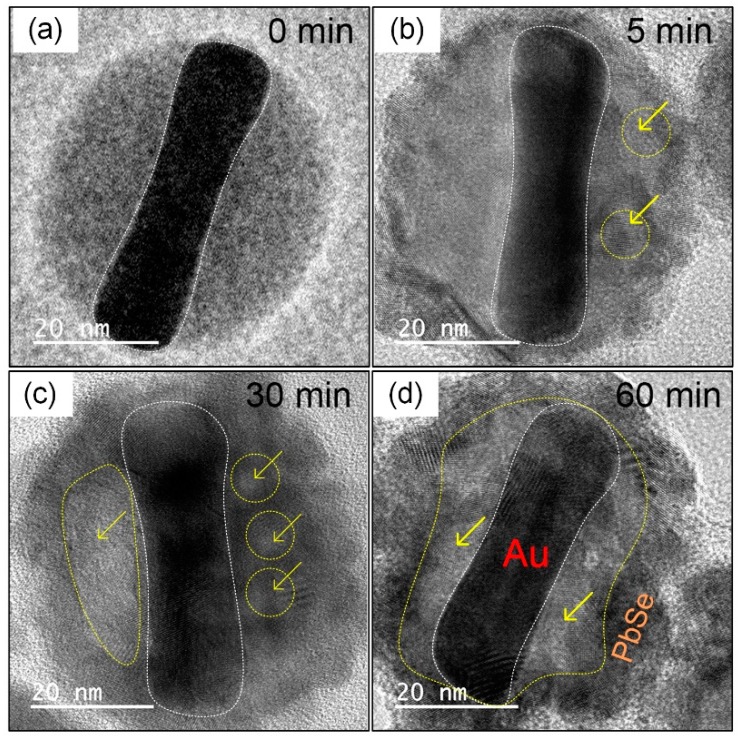
TEM images of (**a**) the initial Au@Se NRs and the products obtained at different reaction times of (**b**) 5 min, (**c**) 30 min and (**d**) 60 min for synthesis of Au–PbSe yolk-shell NRs. The yellow arrows indicate the positions of nanovoids.

**Figure 4 nanomaterials-10-00564-f004:**
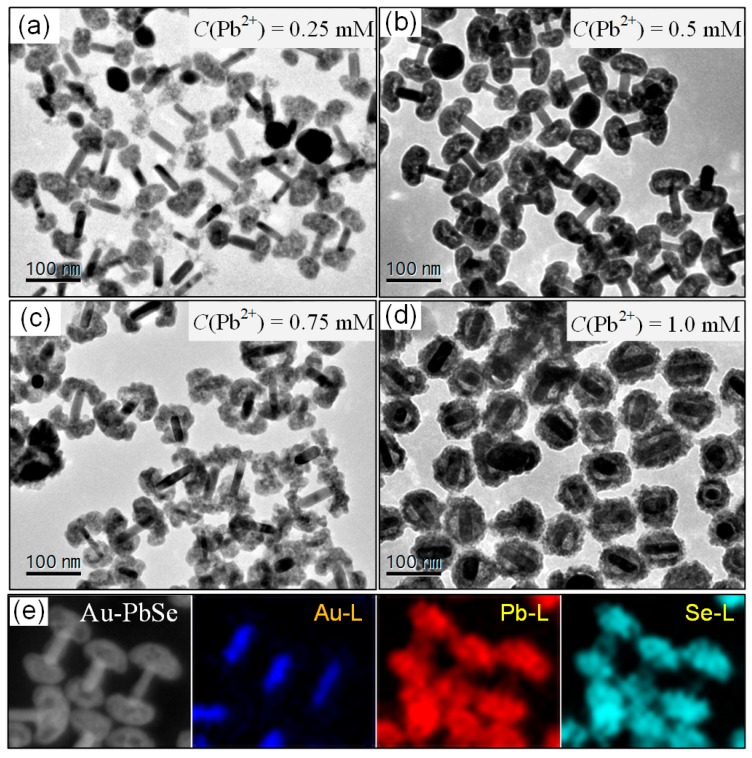
Used Au@Se NRs with 13 nm shells as seeds, the corresponding TEM images of four parallel samples with the initial molar concentration of Pb^2+^ ions of (**a**) 0.25 mM; (**b**) 0.50 mM; (**c**) 0.75 mM and (**d**) 1.0 mM, respectively. (**e**) The corresponding HAADF-STEM image and EDS analysis of Au–PbSe nanodumbbells.

**Figure 5 nanomaterials-10-00564-f005:**
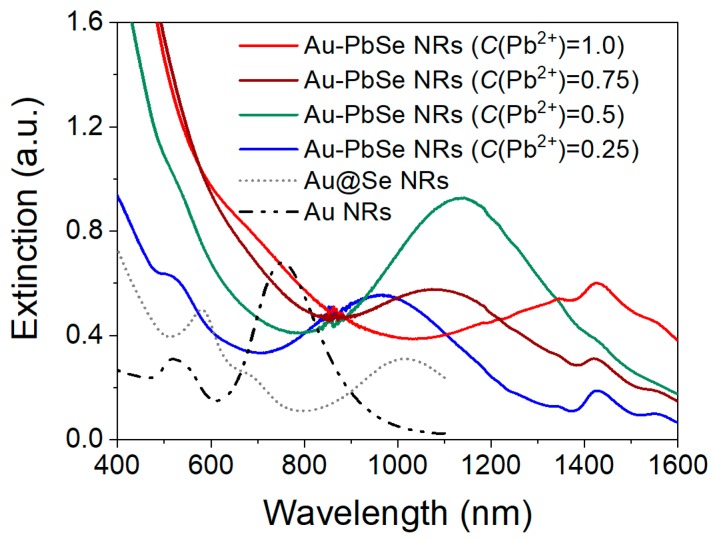
Extinction spectra of original Au NRs, Au@Se NRs with 13 nm shells, and the corresponding four parallel samples obtained at different initial Pb^2+^ concentrations of 0.25 mM, 0.5 mM, 0.75 mM and 1.0 mM.

**Figure 6 nanomaterials-10-00564-f006:**
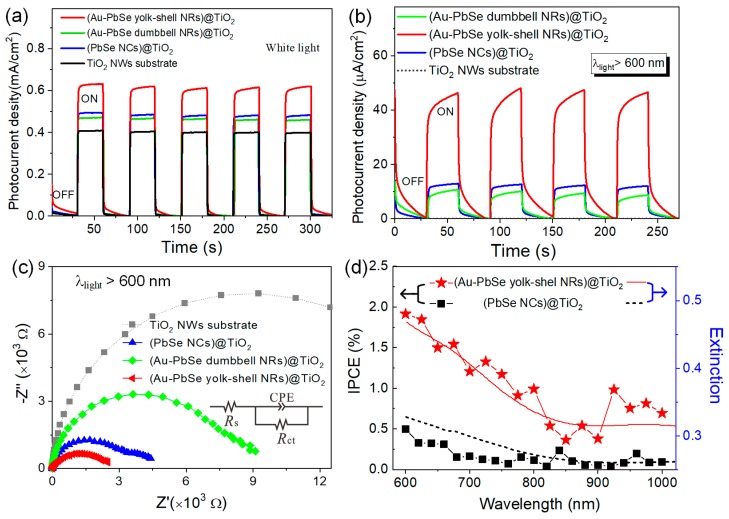
Photocurrent experiments for TiO_2_ nanowires (NWs) film photoanodes sensitized with PbSe NCs, Au–PbSe yolk-shell and dumbbell NRs, respectively. The data were collected at an applied potential of 0.1 V *vs* Ag/AgCl: (**a**) under white light and (**b**) light illumination of *λ* > 600 nm; (**c**) Corresponding EIS spectra for four parallel photoanodes with light illumination of *λ* > 600 nm; (**d**) Incident photon to current conversion efficiency (IPCE) plots of (PbSe NCs)@TiO_2_ and (Au–PbSe yolk-shell NRs)@TiO_2_ in the wavelength range 600–1000 nm. The extinction spectra of PbSe and Au–PbSe yolk-shell NRs sensitizers are also shown.

**Figure 7 nanomaterials-10-00564-f007:**
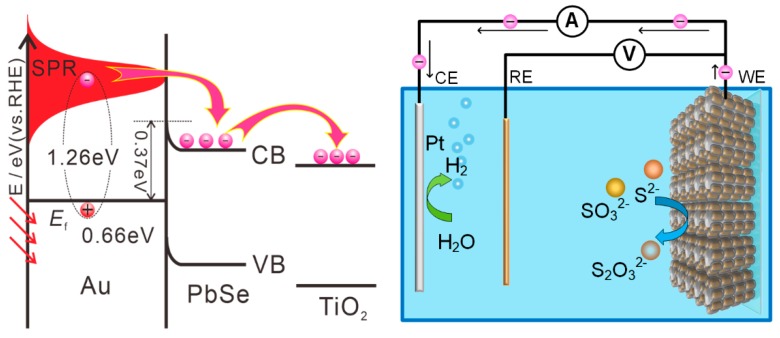
Schematic illustration of device configuration and proposed energy band structure mechanism of TiO_2_ photoelectrode sensitized with Au–PbSe hybrid NRs.

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
