# Peer review of "Controlled Synthesis of Au Nanocrystals-Metal Selenide Hybrid Nanostructures toward Plasmon-Enhanced Photoelectrochemical Energy Conversion"

_nanomaterials, 2020, doi:10.3390/nano10030564_

Round 1

Reviewer 1 Report

See the attachment for the report.

Reviewer 2 Report

This is an experimental paper dealing with synthesis of some complex metal-semiconductor nanomaterial systems and their performance in light harvesting.

The system chosen for the study is a two-component nanostructure of Au NR and a metal selenide. The selenide morphology may be altered to produce dumbbell like structure where the selenide is selectively deposited at the two ends of the Au NR, or a more uniform deposition of the selenide on the NR to give a yolk-shell structure.

The synthesis protocol is quite complex. The TEM investigations of the growth and development of this morphology and chemistry are interesting and worth publishing.

The photo response of these materials obviously show enhancements by the selenide presence. The effect of the different morphologies dumbbell and yolk-shell are also clear. However, the explanations of physics given could be worded better. The differences between interface transfer resistances are striking in the EIS spectra.

Below are some specific comments to authors to improve the manuscript.

  1. The FDTD simulation of SPR enhancement does not make significant contribution to the work. This part could be moved to the Supporting Information without affecting the scientific rigor of the paper. The local field distributions shown in Fig 5 (c) and (d) also could be moved to SI. Basically, the link between SPR and photo activity of these structures appears to be speculative and hence the authors may want to think about down playing this link. There is definitely, enhanced photoelectric activity, and there is possible SPR activity also by the simulation. Whether they are linked is speculative.
  2. Use of the term “hollow” shells in Section 3 is misleading. There are voids in the structure but the interior part is solid Au; also the voids do not play any part in the photo activity of the structures. Therefore, the voids are not important for the function. Use the terms dumbbell and yolk-shell to describe the structures but emphasize the presence of voids.
  3. In Fig 3, I would strongly recommend labelling the Au, selenide, the void for clarity.
  4. Explain why you chose to use λ > 600 nm light in Section 3.4.
  5. Page 11, lines 397-400. The sentence explaining the physics in the dumbbell structure is confusing. Could you re-phrase it?
  6. Page 11, line 410. The SPR absorption spectrum referred to is the calculated one, not the experimental one?

Reviewer 3 Report

Controlled Synthesis of (Au Nanocrystals)-(Metal Selenide) Hollow Hybrid Nanostructures toward Plasmon-Enhanced Photoelectrochemical Energy

Conversion

By Ling Tang et al.

This work shows some strategies to process hybrid nanostructures consisting of Au-NRs-PbSe. There are some issues the authors must address before the paper can be reconsidered for publication in “Nanomaterials”

  1. The abstract must be rewritten in clear, concise and short sentences, removing superlatives.
  2. Line 53: remove “to receive”, replace with “for”
  3. Line 68: remove “with their anisotropic morphology” replace with “depending on the dimensions of the NRs”
  4. Line 70: replace “which “ with “and”
  5. Line 71: remove “charming”. A structure cannot be “charming” but a person can!
  6. Line 78: replace “maneuvering” with “tuning”
  7. Lines 78-80: sentence starting with “even so…” remove or elaborate on this topic
  8. Line 81: What is meant by “controllable symmetries”? Explain.
  9. Line 126: The structure and properties of these hybrid NRs are not reported in the paper. Remove the whole 2.5. section
  10. Line 179: FDTD calculations: Provide more details about the calculations, e.g. the dielectric functions used (with corresponding references) and the boundary conditions
  11. Lines 200-205: The selenide ions are neutralized on the Au-surface. Whre do the 2 electrons come from? what happens when the surface of Au is occupied with the first Se layer?

It is important to write the electrochemical reactions!

  1. Line 220: What is meant by “12 nm hollow”. The sentence is not clear. Rewrite
  2. Lines 247-248: It is claimed that the Se shell is uniform, but it is only uniform in the transverse direction. Correct and explain.
  3. Lines 250-259: Are these (Fig. 3) in situ TEM images? Obviously not! Then These are only speculations, and the so-called voids are probably artefacts. You have a reaction at the nanoscale: the formation of PbSe involves diffusion of Se atoms into the solution and their replacement with Pb atoms. Kirkendall is probably not the mechanism operating here. Either corroborate or correct!
  4. 3 caption: remove “point” replace with “time”
  5. Optical properties: The spectrum of Au-Se shows a distinctive shoulder close to 700 nm. Explain. The green curve shows a higher absorption in the NIR region than the other nanostructures. Explain! Was the concentration the same for all the samples? the green, red and magenta curves are devoid of the transvers peak. Explain.
  6. 5: The extinction spectra of Au and Au-Se terminate at 1100 nm while the other structures terminate at 1600 nm. Explain. For Au-PbSe (0.5, 0.75 and 1.0) there is a small peak at about 1450 nm. The authors claim that this is a red-shifted L-LSPR. I think that this is an artefact of the measurement. Rather the L-LSPR is somewhere between 1000 and 1100 nm. The T-SPR is completely damped by PbSe. Against which reference were the measurements done?
  7. The whole description and discussion of the optical properties is confusing and lacks scientific rigor. Correct.
  8. 6: is the yolk-shell structure investigated in these experiment for 0.75 or 1.0 mM Pb2+?

in either case, the plasmonic absorption is rather low, and in the case of 1mM Pb2+ there is only a somewhat intense background. So how can the authors explain their photoelectrochemical results which show “enhanced photoelectrochemical properties” for these structures? We would rather expect the dumbbell structures to have a higher performance. Explain

  1. What is exactly the wave length used (l>600 nm is not really convincing)
  2. Line 450: remove the structures which have not been investigated in the paper!

Round 2

Reviewer 1 Report

The authors have properly revised the manuscript. It is recommended for publication.

Reviewer 3 Report

The authors failed to answer my questions satisfactorily.

Response 14: I am still not convinced by the mechanism advanced, and the so-called voids, if any, may be due to electron beam effects. As already pointed out, we have diffusion at the nanoscale with Se diffusing out and the same amount of Pb diffusing in. Why should Kirkendall voids form? The authors do not provide a sound rationale for their claims.

Provide reaction equations in the manuscript, citing relevant references.

Response 17: the authors fall into contradiction. On the one hand they admit that the sharp peak at 1450nm is an artifact on the other, however, they say that the peak at 1450 nm is the shifted LSPR!

Response 19 is not convincing either. How can plasmon enhancement happen if there is no distinct plasmon peak?

The English is still very shaky. Extensive language editing is mandatory.

Replace in all Fig captions "magnitude" by "magnification" and so on..

Round 3

Reviewer 3 Report

The manuscript can now be accepted.